# Endoscopic Ultrasound-Guided Radiofrequency Ablation as an Future Alternative to Pancreatectomy for Pancreatic Metastases from Renal Cell Carcinoma: A Prospective Study

**DOI:** 10.3390/cancers13215267

**Published:** 2021-10-20

**Authors:** Brice Chanez, Fabrice Caillol, Jean-Philippe Ratone, Christian Pesenti, Philippe Rochigneux, Géraldine Pignot, Jeanne Thomassin, Serge Brunelle, Jochen Walz, Naji Salem, Marc Giovannini, Gwenaelle Gravis

**Affiliations:** 1Department of Medical Oncology, Institut Paoli-Calmettes, 13009 Marseille, France; chanezb@ipc.unicancer.fr (B.C.); rochigneuxp@ipc.unicancer.fr (P.R.); 2Centre de Recherche en Cancérologie de Marseille, INSERM UMR1068, CNRS UMR7258, Institut Paoli-Calmettes, 13009 Marseille, France; 3Aix Marseille Université, 13010 Marseille, France; 4Department of Gastroenterology and Endoscopy, Institut Paoli-Calmettes, 13009 Marseille, France; caillolf@ipc.unicancer.fr (F.C.); ratonej@ipc.unicancer.fr (J.-P.R.); pesentic@ipc.unicancer.fr (C.P.); giovanninim@ipc.unicancer.fr (M.G.); 5Department of Urologic Surgery, Institut Paoli-Calmettes, 13009 Marseille, France; pignotg@ipc.unicancer.fr (G.P.); walzj@ipc.unicancer.fr (J.W.); 6Department of Pathology, Institut Paoli-Calmettes, 13009 Marseille, France; thomassinj@ipc.unicancer.fr; 7Department of Radiology, Institut Paoli-Calmettes, 13009 Marseille, France; brunelles@ipc.unicancer.fr; 8Department of Radiotherapy, Institut Paoli-Calmettes, 13009 Marseille, France; salemn@ipc.unicancer.fr

**Keywords:** glandular metastases, endoscopy, radiofrequency ablation, renal cell carcinoma, pancreatic metastases

## Abstract

**Simple Summary:**

Glandular metastases and more precisely pancreatic metastases (PM) from renal cell carcinoma (mRCC) are associated with a long survival. Focal treatment in order to control oligo-metastatic disease and avoid systemic therapy is a standard in RCC. However, pancreatic radiofrequency ablation remains a marginal and under evaluated technic. Standard treatment remains pancreatectomy with hazardous outcomes. We report here the largest series of endoscopic radiofrequency ablation (EUS-RFA) on PM for patient treated at Institut Paoli-Calmettes for a mRCC. Patients with progressive PM was treated under general anesthesia with an ultrasound guided endoscopic radiofrequency ablation procedure then followed by CT-scan. We prospectively included 12 patients, median age was 70.5 years old and median size and number of PM at inclusion was 17 mm and 1 respectively. All 26 procedures performed went as planned and no immediate complications were experienced. After 27.7 months of follow-up, the 6- and 12-month focal control rates were 84% and 73% respectively. Two severe complications occurred after EUS-RFA but were totally resolved after specific treatment. In total, EUS-RFA is feasible and displayed an excellent focal control and should be integrated in the arsenal to treat PM from mRCC in order to spare systemic therapy and/or pancreatic surgery.

**Abstract:**

Background: Pancreatic metastases (PM) from renal cell carcinoma (RCC) are rare, are associated with favorable outcomes and are usually handled by surgery or VEGFR inhibitors, which both have side effects. Endoscopic Ultrasound (EUS)-guided radiofrequency ablation (RFA) is an innovative approach to treat focally deep metastases and could be a relevant technique to control PM from RCC. Methods: This monocentric, prospective study aimed to evaluate the safety and efficacy of EUS-RFA to treat PM. We included patients with confirmed and progressive PM from RCC. PM was ablated under general anesthesia with a linear EUS scope and a EUS-RFA 19-gauge needle electrode placed into the tumor. Results: Twelve patients from Paoli-Calmettes Institute were recruited between May 2017 and December 2019. Median age was 70.5 years (range 61–75), 50% were female, 100% were ECOG 0–1. At inclusion, mean PM size was 17 mm (range 3–35 mm); and all were progressive before EUS-RFA. Seven patients had EUS-RFA as the only treatment for RCC. We performed 26 EUS-RFA procedures and 21 PM was ablated. Median follow up was 27.7 months (range 6.4–57.1). For evaluable PM, the 6- and 12-month focal control rates were 84% and 73% respectively. One patient treated with TKI developed a paraduodenal abscess 2 months after EUS-RFA and another patient with biliary stent developed hepatic abscesses few days after EUS-RFA. No other severe side effects were experienced. Conclusions: in this series, which is the largest ever reported, we showed that EUS-RFA is feasible and yields an excellent local control rate for PM from mRCC. With manageable complications, it could be a valuable alternative to pancreatic surgery in well-selected patients.

## 1. Introduction

Pancreatic metastases (PM) are rare, but when detected, renal cell carcinoma is the most frequent primary tumor [1]. PMs are detected in 1 to 6% of metastatic renal cell carcinoma (mRCC) and are frequently associated with other glandular metastases [2], which are known to confer a better outcome compared to liver, bones or brain metastases [3,4,5]. Although molecular mechanisms associated with long survival of patients presenting PM are poorly understood, primary tumors exhibit slow-growing characteristics (low Fuhrman/ISUP grade, long period between primary tumor and metastastic relapse) [1] or specific molecular/immune features (linear and monogenic primary to metastasis evolution, reduced effector T cell gene signatures and increased angiogenesis dependence) [6,7].

Focal therapies are now a standard treatment of oligometastic mRCC [8]. Pancreatic resection is the most common option to control isolated PM and to avoid compressive symptoms, whereas tyrosine kinase inhibitors (TKI) are preferentially used when other metastatic sites are detected [9,10].

However, these strategies present morbid side-effects and not all patients are fit enough to undergo pancreatectomy [1]. Pancreatic surgery for RCC with solitary PM has been extensively reported and 5-year survival rates range 50 to 75% [1,10,11,12]. However, this complex surgery has a post-operative mortality of around 2% and morbidity varies from 13.5 to 43% (pancreatic fistula, infections or diabetes) [9,13,14]. Thus, less invasive techniques need to be developed and validated.

Endoscopic Ultra-Sound (EUS)-guided radiofrequency ablation (RFA) is a micro-invasive technique which need a classical therapeutic EUS scope (with 4 mm working channel) and a dedicated 19 Gauge EUS-RFA needle [15]. The EUS-RFA probe delivers locally a high-frequency (400–500 kHz) alternating (sinusoidal) current that generates electromagnetic high-temperature cell damages, leading to apoptosis and coagulative necrosis, limiting bleeding [16]. Percutaneous-RFA has shown efficacy to treat both primary and secondary liver tumors and recent reports showed successful transposition of this technique to pancreatic tumors [15,17]. The technique seems promising in hyper-vascularized tumors like RCC. However, large cohorts of RFA (external or endoscopic) in the treatment of PM are lacking and only case reports are available displaying heterogeneous results and toxicity profiles [18,19].

We report here the efficacy and safety of EUS-RFA in a series of patients with PM from RCC treated at the Institut Paoli-Calmettes (Comprehensive Cancer Center, Marseille, France).

## 2. Materials and Methods

This prospective, monocentric, non-controlled, single arm pilot study aimed to analyze the safety and the focal control of EUS-RFA used to treat PM from RCC.

### 2.1. Patients

Eligible patients was >18 years-old, had metastatic clear cell RCC with progressive but limited PM (<4 PM, maximum diameter: 25 mm) [17], ECOG 0–1, eligible to general anesthesia and treated or not for extra-PM disease. Extra PM metastases, when present, had to be limited and stable. The indications of EUS-RFA were validated during cancer multidisciplinary team meetings and all patients gave their written informed consent. The protocol was approved by the institutional review board of Institut Paoli-Calmettes (the IRB number of this study is: PRAM-1-IPC 2018-013, approval date: 4 July 2018).

### 2.2. EUS-RFA Procedure

EUS-RFA was performed under general anesthesia, with a linear EUS scope.

We used an EUS-RFA system (STARmed, Seoul, Korea) consisting of a 19-gauge needle electrodes covered with a sheath, except for the terminal 1 cm with a sharp conical tip for energy delivery, and an radiofrequency (RF) generator. This system features an internal cooling system which circulates a chilled saline solution through the needle electrode during the RFA procedure to prevent charring on the surface of the electrode and to improve the accuracy of the ablation. The lesion was punctured at 1 to 3 different points, and 50 W ablation power was delivered at the different sites under direct EUS control. Ablation time, related to tissue impedance measured in real time by the RF generator, was short (10–20 s but when PM was >20 mm, time was slightly longer from 30–60 s) (Appendix A). Bioimpedance reflects the property of a tissue to conduct electricity. An impedance rise is an indirect sign of cellular disruption, tissue charring, and hampering of electrical conduction. When a sudden rise in impedance was registered, power delivery was stopped.

### 2.3. Prophylaxis and Evaluation of Immediate Complications

Before ablation, all patients received non-steroidal anti-inflammatory treatment (indomethacin by rectal route) to prevent acute pancreatitis and antibioprophylaxis using cefotaxime.

Patients were kept two days in the hospital for surveillance of pain, bleeding, perforation signs, fever or other acute pancreatitis symptoms.

Before general anesthesia, indomethacin and cefotaxime were administrated to prevent acute pancreatitis. EUS-RFA was performed with a linear EUS scope and a EUS-RFA system (STARmed, Seoul, Korea—19-gauge needle electrode). The lesion was punctured (at 1–3 points) (Appendix A), ablation time was short (50 watt–30 s to 1 min) and controlled by tissue impedance measured in real time by the RF generator.

### 2.4. Tumor Response Assessment

CT-scan was performed 2 months after each EUS-RFA procedure, then every 3 to 6 months for follow-up. Response was assessed for each treated PM using size and contrast uptake (Figure 1). Complete response was defined by complete disappearance of the lesion or absence of any contrast uptake at the arterial phase [17]; partial response was a regression of 30% or more of the tumor contrast uptake, stable disease was defined with size change comprised between −30% and +20% of the contrast uptake of PM. Progressive disease was defined as a >20% increase in size of PM with contrast uptake or new compressive symptoms [20].

Focal control rate was defined as the sum of complete response, partial response and stable disease rates.

## 3. Results

Between June 2017 and December 2019, twelve patients underwent EUS-RFA for PM from mRCC, at the Institut Paoli-Calmettes (Marseille, France), in the Endoscopic and Interventional unit.

All had undergone past nephrectomy for RCC, 50% were female, and the median age at inclusion was 70.5 years-old (range 62–75). Median time between nephrectomy and PM was 13.6 years (range 0–22.2) (Table 1). For 58%, PM remained the unique metastatic site until the last evaluation. The mean size of PM was 17 mm (range 3–35) and 33% had an initial size >20 mm (range 22–35) (Appendix A). One patient developed jaundice and required an endoscopic drain with a biliary stent.

Five patients (42%) had received a systemic TKI before the first EUS-RFA (Table 1).

Three patients began a TKI new line after EUS-RFA, all for extra pancreatic progression.

We performed 26 EUS-RFA sessions to treat a total of 21 PMs (Appendix A). Nine patients underwent ≥2 EUS-RFA due to an incomplete necrosis of PM at first evaluation (Appendix A).

A second and third session for a same patient could be motivated by (1) the presence of untreated metastases in the first session (due to extremely low size or new lesion); (2) uncomplete PM necrosis evaluated by CT-Scan defined by contrast enhanced remnant tissue.

### 3.1. Toxicities

All procedures achieved technical success with no immediate complication and all patients spent 2 days at hospital for each session.

Two patients experienced complications requiring hospitalization and intervention (grade IIIb Clavien-Dindo [21]). In the first patient, treated by TKI at the time of EUS-RFA, a duodenal abscess occurred two months after the 2nd EUS-RFA. In the second patient, the only one with a biliary stent, a symptomatic hepatic abscess occurred one week after the EUS-RFA. At the time of analysis, all adverse effects had been resolved, and no further complication was detected in any patient.

These two patients were the only ones in the cohort to continue TKI during EUS-RFA and to have a biliary stent. These observations had to be confirmed in a larger population, but patients treated with EUS-RFA should probably stop TKI and not have biliary stents before the procedure.

### 3.2. Focal Control and Survival

Median follow-up since the first EUS-RFA was 27.7 months (range 6.4–57.1).

Among the 21 PMs treated, 40% had a complete response at 12 months. The 6-month and 12-month focal control rates were 84.2% and 73.3% respectively among evaluable PMs. Progressive disease was observed in only 15.8% and 26.7% of evaluable PMs at 6 and 12 months respectively (Table 2). All patients were alive at the time of analysis and median progression free survival for PM was 25.38 months (Figure 2).

Comparing PM > 20 mm to those <20 mm, the 6-month focal control rate was lower (92% vs. 71%) (Appendix A) and the time to progression was shorter although the difference was not statistically significant (*p* = 0.06) (Appendix A). Tumor size was not significantly associated with a better response (OR 0.37, *p* = 0.43) in logistic regression analysis (Appendix A).

After the last EUS-RFA, two patients received a new systemic therapy, one performed duodeno-pancreatectomy for PM (the same patient who developed paraduodenal abcess), and 4 patients underwent focal treatment of other RCC metastases (metastasectomy and/or RFA on the liver or lung). Focal directed therapies for oligometastatic disease are considered as a standard treatment as they improve both overall survival and cancer-specific survival in RCC [8,22,23]. Here, EUS-RFA was perfectly integrated in a multimodal focal therapy strategy, which is of high importance for this ‘slow growing tumor’ population.

## 4. Discussion

### 4.1. EUS-RFA for Managing Slow Growing Metastases and Avoiding Morbid Surgical Resection

In this study, among the 26 procedures performed, only two were associated with severe side effects in 2/12 patients (17%). Two teams published case reports of external RFA for RCC-PM with contrasted results: for Elias et al., the two patients treated with RFA during laparotomy or laporoscopy to treat 2 to 3 PM ≤ 20 mm, complementary to the resection of other RCC metastases, both experienced severe pancreatitis leading to emergency surgery. The authors concluded that “in 2003, RFA is not yet usable and safe” [18]. However, for Carrafiello et al., a percutaneous RFA for a 20 mm PM was performed safely, resulting in no residual disease one year later [19].

In our series, no acute pancreatitis was observed with the systematic administration of indomethacin and cefotaxime in primary prevention.

Our data highlighted that EUS-RFA can achieve a good focal control of PM with only 15.8 and 26.7% of progression at 6 and 12 months. According to these results, EUS-RFA should be a key option to treat and control PM from RCC.

Finally, all the 4 patients that experienced PM progression after EUS-RFA underwent additional treatment including immune check-point inhibitors (ICI). Further data could also address whether EUS-RFA could trigger an antigenic reaction [16,24] and bypass the PM resistance mechanism to ICI [7].

The main limitation of our work is the low number of treated patients. However, with 26 procedures, this study provided relevant information about the good tolerance and performance of EUS-RFA for PM.

### 4.2. RFA and Immunotherapy

Focal treatment like radiation can trigger inflammatory cell activation, pictured in hepatocellular carcinoma (HCC), in which RFA is a standard treatment, and where studies observed a T cell response against tumor tissue and an increased number of systemic NK cells and cytotoxicity [25,26]. Recent immune monitoring in 80 patients with HCC < 5 cm reported that systemic variations of activated NK cells (NKp30+) after RFA was associated with tumor recurrence and with increased systemic CD8 Central Memory [24]. This overall activation of the immune system by RFA could be even more efficient in association with immune checkpoint inhibitors, which are now a standard of care for mRCC [27,28,29]. PM was recently proposed as a RCC metastatic localization associated with low response and survival with nivolumab [7]. Such an ablation with EUS-RFA could trigger an antigenic reaction and sensitize PMs to immune checkpoint inhibitors.

### 4.3. Study Limitations

Obviously, the low number of treated patients and the relatively short follow-up are the main limitations of our study, but with 26 procedures, this study provided relevant information about the good tolerance and performance of EUS-RFA for PM. Longer follow up will also provide more mature data, especially among tumors what are not evaluable (Table 2 and Appendix A).

However, a larger comparative study might be hard to set-up because of the rarity of PM.

Furthermore, data from primary pancreatic tumors (adenocarcinomas or PNETS) reinforce the level of evidence of good tolerance and efficacy of this innovative procedure [17,30].

## 5. Conclusions

Even if further prospective data are needed to reinforce our findings, EUS-RFA to treat PM from RCC appears to be implemented as a promising option in the armamentarium of focal therapy in well selected patients.

## Figures and Tables

**Figure 1 cancers-13-05267-f001:**
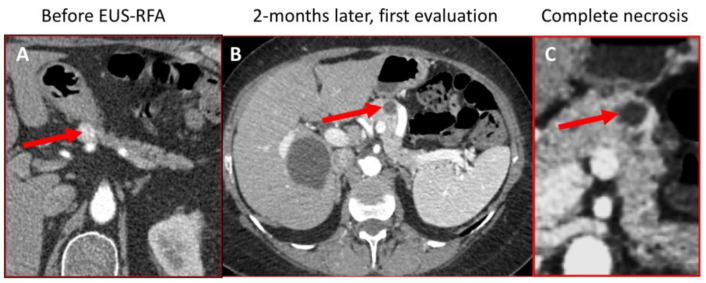
Following contrast enhancing PM with CT-scan before and after EUS-RFA. Abdomen CT-Scan with contrast injection at artery phase, before (**A**) and after endoscopic radiofrequency ablation of PM in the body of the pancreas, displaying complete response without contrast enhancement in RFA scar ((**B**) and at magnification (**C**)). PM: pancreatic metastases, CT: computed tomography, EUS-RFA: Endoscopic Ultra-Sound guided RadioFrequency Ablation.

**Figure 2 cancers-13-05267-f002:**
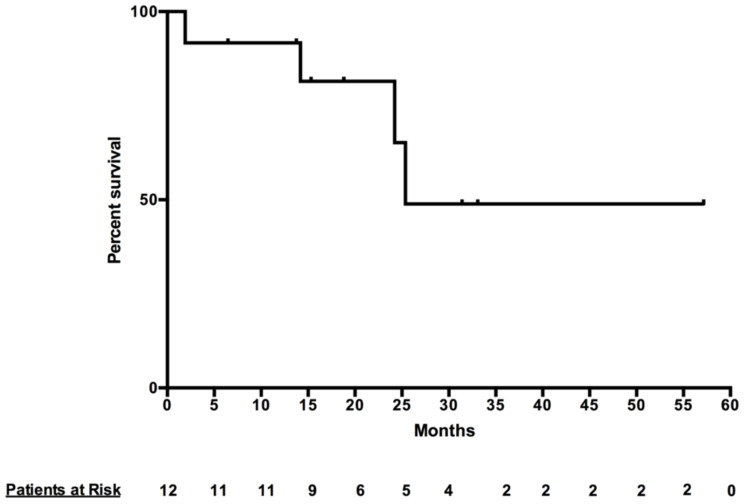
Patient Progression Free Survival.

**Table 1 cancers-13-05267-t001:** Baseline patients’ characteristics (*n* = 12).

Baseline Patients’ Characteristics (*n* = 12)			*n*	%
Left Kidney		*n* (%)	6	60
Gender		Female *n* (%)	6	50
TNM		T1/T2	9	75
		T3/T4	1	8
		Unknown	2	17
Fuhrman Grade		I/II	7	58
		III/IV	3	25
		Unknown	2	17
IMDC		Good/Intermediate	4/5	33/42
		Unknown	3	25
Systemic Treatment	Before EUS-RFA	VEGFR-TKI/ICI	4/0	33
	During EUS-RFA	VEGFR-TKI/ICI	1/1	8
	After EUS-RFA	VEGFR-TKI/ICI	3/0	25
	No systemic treatment		7	58
Other Metastatic Sites During RCC Evolution		None	7	58
		Lung	5	42
		Brain	1	8
		Liver	2	17
		Thyroid	1	8
		Others	2	17
Median Time from Nephrectomy to 1st Metastasis		Years (range)	11.6	(0–22.2)
Median Time from Nephrectomy to PM		Years (range)	13.6	(0–22.2)
Pancreas as the 1st Metastatic Site		*n* (%)	9	75
Symptoms at PM Presentation		None	9	75
		Jaundice	1	8
		Pain	2	17
PM Localization at Presentation		Head	7	33
		Uncus	4	19
		Body	6	29
		Tail	4	19
Median PM SIZE at Presentation		Millimeters, (range)	17	(3–35)
Median Number of PM at Presentation (range)		1	(1–4)	
PM Number at Presentation (*n*, %)		1 2 3–4	58 17 25	

TNM: Tumor Node Metastases Classification, IMDC: International Metastatic RCC Database Consortium International, EUS-RFA: Endoscopic Ultra-Sound guided Radio Frequency Ablation, VEGFR-TKI: Vascular Endothelial Growth Factor Receptor-Tyrosine Kinase Inhibitor, ICI: Immune Check Pont Inhibitor, RCC: Renal Cell Carcinoma, PM: pancreatic metastases.

**Table 2 cancers-13-05267-t002:** Objective response rates at 2, 6 and 12 for PM.

Time Point (Evaluable Lesion)	CR (%)	PR (%)	SD (%)	PD (%)	Focal Control Rate (%)	N/E
2-months (*n* = 21)	33.3	14.3	42.9	9.5	90.5	0
6-months (*n* = 19)	26.3	31.6	26.3	15.8	84.2	9.5
12-months (*n* = 15)	40	33.3	0	26.7	73.3	28.6

PM: pancreatic metastasis CR: complete response, PR: partial response, SD: stable disease PD: progression disease; N/E Not evaluable lesion (that have progressed or follow up is < to the evaluation time point); Focal Control R ate = CR + PR + SD.

## Data Availability

The data presented in this study are available on request from the corresponding author. The data was generated from clinical history of patients therefore they are not publicly available.

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
