# Peer review of "Endoscopic Ultrasound-Guided Radiofrequency Ablation as an Future Alternative to Pancreatectomy for Pancreatic Metastases from Renal Cell Carcinoma: A Prospective Study"

_cancers, 2021, doi:10.3390/cancers13215267_

Round 1

Reviewer 1 Report

Thank you for making corrections based on the reviewer's comments.
Although sufficient modifications have been made, I think that the title and abstract need further corrections. 

Although EUS-RFA is regarded as an possible thereapeutic options, the title seems to suggest it as a standardized treatment. It is desirable to add expressions such as potential, possible, etc. to the title.

The closing sentence of the abstract "with few major complications" should be revised.

Thank you

Author Response

Dear Reviewer,

We would like to thank you for your suggestions and review of our manuscript cancers-1394185 and we have made the modification to the original manuscript as follow :

Comment 1 : Although EUS-RFA is regarded as an possible thereapeutic options, the title seems to suggest it as a standardized treatment. It is desirable to add expressions such as potential, possible, etc. to the title.

Thank you for this suggestion. We have then modified the title as follow (line 2-4) ‘Endoscopic Ultrasound-Guided Radiofrequency Ablation as an Future Alternative to Pancreatectomy for Pancreatic Metastases from Renal Cell Carcinoma: A Prospective Study’

Comment 2 : The closing sentence of the abstract "with few major complications" should be revised.

 As suggested by the reviewer, we revised the closing sentence of the abstract (line 48-49) ‘With manageable complications, it could be a valuable alternative to pancreatic surgery, in well-selected patients.’

Reviewer 2 Report

The authors addressed most of the comments of the prior reviewers. Below a few additional issues for the authors to consider:

-If possible, please provide a reference for your definitions of complete and partial response.

-The conclusions statement is a bit overreaching. Please change 'practice changing technique' to 'promising technique'

-abstract: change 'pancreatic radiofrequency' to 'pancreatic radiofrequency ablation'

Author Response

Dear Reviewer,

Thank you for your comments and suggestions for our research paper N°. cancers-1394185 and we have made the modification of the original manuscript as follow:

Comment 1 : if possible, please provide a reference for your definitions of complete and partial response.

We thank the reviewer for this request and we had two references:

- the RECIST 1.1 consensus because we analysed the size of the treated PM using RECIST (Line 130) - [21] 10.1016/j.ejca.2008.10.026

- but also, the contrast uptake was used when the PM wasn't enhanced by contrast product revealing a complete necrosis, as describe by Barthet et al, in another EUS-RFA report in NET (line 127) : [18] 10.1055/a-0824-7067

“Complete response was defined by complete disappearance of the lesion or absence of any contrast uptake at the arterial phase [18]; partial response was a regression of 30% or more of the tumor contrast uptake or size change comprised between -30% and +20% of the contrast uptake of PM. Progressive disease was defined as a >20% increase in size of PM with contrast uptake or new compressive symptoms [21] »

Comment 2 -The conclusions statement is a bit overreaching. Please change 'practice changing technique' to 'promising technique'

We agree with this suggestion and have changed as follow (line 242-244) “Even if further prospective data are needed to reinforce our findings, EUS-RFA to treat PM from RCC appears to be implemented as a promising option in the armamentarium of focal therapy, in well selected patients.”

Comment 3 -abstract: change 'pancreatic radiofrequency' to 'pancreatic radiofrequency ablation'

We have changed line 20 : “[…] pancreatic radiofrequency ablation remains a marginal and under evaluated technic. »

This manuscript is a resubmission of an earlier submission. The following is a list of the peer review reports and author responses from that submission.

Round 1

Reviewer 1 Report

A very intersting pilot study on safety and efficacy of EUS-RFA for treatment of pancreatic metastases from renal cell carcinoma which could offer a relatively safer alternative to surgery in these patients

Major comments

  1. 12 patients were treated over 26 sessions for ablating 21 PM. Which were the criteria for deciding for a second or third session. Uncomplete necrosis was evaluated by CT Scan? Any suspiscion of remnant tisuue was retreated?
  2. 7 patients had EUS RFA as single treatment and 5 received systemic TKI before. How were there selected? Was the outcome different between both group?
  3. Complete response and focal control rate should be defined
  4. EUS-RFA has been done in a highly expert center. Some détails should be provided on the technique of ablation (aside power and duration), including needle positioning, visualization during ablation, possible decision criteria fo a second or third ablation of the PM during the same procedure. One or two illustrative videos as supplementary material would be welcomed

Reviewer 2 Report

I am very pleased to review an interesting study about EUS guided RFA for pancreatic RCC metastasis, I think it is worthy to give an evidence of clinical application of EUS guided RFA for these patients . however, there are some major concerns, thus I'd like to ask the author to revise major revision points.

Major revision

1. For prospective studies, the IRB number must be presented. And because it is a small sample study, you'd better label it as a single arm pilot study. 

2. Two patients had procedure related complications, and one patient had pancreaticoduodenectomy after EUS-RFA. because PPPD  and RFA related complications occurred in 3 out of 12 patients, the procedure itself cannot be said to be safe and effective. I think that it is very risky to generalize the positive therapeutic effect of EUS-RFA for pancreatic RCC metastasis based on the experiences of only 12 patients.  So I think you should present the possibility of a therapeutic effect in a weaker expression than the current phrase. 

3. As your comment, combining TKI with EUS-RFA may increase the risk of bleeding, but it cannot be definitive (only one case). (Rersults = This data has to be confirmed but patients treated with EUS-RFA should stop TKI before and not having biliary stents)….

4. In terms of actual practice of EUS-RFA (EUSRA), 30-50W 15 seconds RFA settings were usually used. But in your article, the ablation time for 30s – 1 min seems to be too long. Besides, because EUS means that the ultrasonic probe and the endoscope are manufactured in one piece, the content of the introduction seems to be wrong. (introduction - Endoscopic Ultrasound (EUS)-guided radiofrequency ablation (RFA) is a micro-invasive technique which combines a classical endoscope, an endoscopic-ultrasonography probe inserted down the working tube and a radiofrequency needle, all placed into the tumor through the digestive wall’)

5. Similarly, the connection between the reference 12/13 and the sentence of the introduction is wrong. EUS-RFA has not proven its effectiveness in liver cancer and pancreatic cancer. (Introduction - EUS-RFA has shown efficacy to treat both primary and secondary liver tumors and recent reports showed successful transposition of this technique to pancreatic tumors 12,13.)

6. Why did you proceed with a size increase of more than 30%? In general, the RECEIST criteria consider 20% or more increase of LD in target lesion as progression, so appropriate reference is needed

7. Since systemic TKI was used in 3 patients after RFA, how the systemic treatment effect would be reflected should be added in your manuscript.

Minor revision

1. For a full understanding, it is necessary to present a photo of EUS RFA procedure.

2. In table 1, median time from nephrectomy to 1st metastasis 11.6 years, but median time from nephrectomy to PM is 13.6 months. Maybe you mean years?

3. In table 1, systemic treatment % , 8/8 is not correct.